# Tuberculosis among refugees and migrant populations: Systematic review

**Abyot Meaza**[1,2]\*, **Habteyes Hailu Tola**[1], **Kirubel Eshetu**[1], **Tedla Mindaye**[3], **Girmay Medhin**[2], **Balako Gumi**[2]

**1** Ethiopian Public Health Institute (EPHI), Addis Ababa, Ethiopia, **2** Aklilu Lemma Institute of Pathobiology, Addis Ababa University, Addis Ababa, Ethiopia, **3** Sibley Memorial Hospital, Johns Hopkins Medicine, Washington, DC, United States of America

\* abimeaza@gmail.com

**Data Availability Statement:** All data and materials that support the final results are presented in the paper and its Supporting Information file.

**Funding:** The author(s) received no specific funding for this work.

## Abstract

Tuberculosis (TB) is an important cause of morbidity and mortality among refugees and migrant populations. These groups are among the most vulnerable populations at increased risk of developing TB. However, there is no systematic review that attempts to summarize TB among refugees and migrant populations. This study aimed to summarize evidence on the magnitude of TB among refugees and migrant populations. The findings of this review will provide evidence to improve TB prevention and control policies in refugees and migrants in refugee camps and in migrant-hosting countries. A systematic search was done to retrieve the articles published from 2014 to 2021 in English language from electronic databases. Key searching terms were used in both free text and Medical Subject Heading (MeSH). Articles which had reported the magnitude of TB among refugees and migrant populations were included in the review. We assessed the risk of bias, and quality of the included studies with a modified version of the Newcastle–Ottawa Scale (NOS). Included studies which had reported incidence or prevalence data were eligible for data synthesis. The results were shown as summary tables. In the present review, more than 3 million refugees and migrants were screened for TB with the data collection period between 1991 and 2017 among the included studies. The incidence and prevalence of TB ranged from 19 to 754 cases per 100,000 population and 18.7 to 535 cases per 100,000 population respectively among the included studies. The current findings show that the most reported countries of origin in TB cases among refugees and migrants were from Asia and Africa; and the incidence and prevalence of TB among refugees and migrant populations is higher than in the host countries. This implies the need to implement and improve TB prevention and control in refugees and migrant populations globally.

**Trial registration**: The protocol of this review was registered on PROSPERO (International prospective register of systematic reviews) with ID number, CRD42020157619.

## Introduction

Tuberculosis (TB) is an infectious disease caused by the bacillus *Mycobacterium tuberculosis* [1]. It typically affects the lungs (pulmonary TB) but can also affect other sites (extra

**Competing interests:** The author(s) declare that there is no any conflict of interest that might influence this work. This does not alter our adherence to PLOS ONE policies on sharing data and materials

pulmonary TB) [2]. TB is causing morbidity and mortality each year among millions across the world. It is the tenth leading cause of death worldwide and the leading cause of a single infectious agent [3]. The recent World Health Organization (WHO) recent report indicated 10 million new cases and 1.4 million deaths occurred globally [3].

Tuberculosis is an important cause of morbidity and mortality among refugees and migrant populations [4]. Refugees are defined as people who are outside their country and cannot return due to a well-founded fear of persecution because of their race, religion, nationality, political opinion or membership in a particular social group [5]. Whereas, migrants are defined as people and/or family members moving to another country or region to better their material or social conditions and improve the prospects for themselves or their family [6].

The number of international migrants reached 244 million in 2015, with almost two-thirds travelling to industrialized countries [7]. The number of refugees also exceeded 70 million in 2018, which was the highest level that the United Nation High Commission for Refugees (UNHCR) has seen in its almost 70 years [8]. Sub-Saharan Africa hosted more than 26% of the world's refugee population, and the number of refugees in the Horn of Africa exceeded 3 million, with Uganda (1.1 million) and Ethiopia (905,000) hosting the largest numbers [9].

Refugees and migrants are the most affected populations at risk of developing TB and other infectious diseases due to their living conditions [7]. Poor living conditions and overcrowding in refugee settlements potentially increase the risk of TB infection [10]. The impact of migration on TB epidemiology is high [11]. The arrival of large groups of refugees can affect TB control efforts in receiving countries by significantly increasing the disease burden and cost of health services [10]. Since the majority of the migrants originate from low-income countries with a high TB burden, it imposes new challenges for global TB control and eradication towards achieving the stop TB strategy [12].

Although there is published evidence and reports in different geographical areas [4, 13–18], there is a lack of systematic review that attempts to summarize the magnitude of TB among refugees and migrant populations. The present systematic review aimed to summarize existing published literature on the magnitude (incidence and prevalence) of TB among refugees and migrant populations living in refugee camps and in migrant-hosting countries.

## Methods

### Search strategy

We conducted a systematic review of published articles and summarized findings about the incidence and prevalence of TB among refugees and migrant populations living in refugee camps and in migrant-hosting countries following PRISMA (Preferred Reporting Items for Systematic Reviews and Meta-Analyses) standards [19] (S1 File).

We systematically searched electronic databases: Medline/PubMed, Web of Science and Google scholar for English language articles. We used sensitive searching method to include all potential articles in the data bases. The search was restricted to papers published in the last 7 years (between 2014 and 2021). The electronic database search was conducted from 01 August 2021 to 4 November 2021. We used a search strategy by combining key-terms: "Tuberculosis", "Multidrug-Resistant", "Pulmonary Tuberculosis", "Pleural Tuberculosis", "Miliary Tuberculosis", "Meningeal Tuberculosis", "Lymph node Tuberculosis", "Laryngeal Tuberculosis", "Mycobacterium tuberculosis", "Prevalence", "Epidemiology", "Cross-Sectional Studies", "Incidence", "Cohort Studies", "mortality", "Refugees", "Refugee Camps", "Migrant" and "Transients and Migrants" both in MeSH and free text terms.

## Inclusion and exclusion criteria

We included cross-sectional studies, retrospective cohort studies and cohort studies which had reported incidence or prevalence as an outcome variable. Eligible studies which were conducted in different parts of the world and published from 2014 to 2021 were included in the review. We excluded studies with different outcome variables, and studies published in non-English languages. The included studies were grouped as incidence and prevalence for the syntheses.

## Study selection process

A total of 1241 records were identified from data base searches and 1188 articles were excluded after title screening due to outcome variable difference with the present review objective. Upon title and abstract screening of 53 articles, 31 articles were excluded based on the exclusion criteria. Then the remaining 22 articles in full text were reviewed for eligibility and 12 articles were excluded. In the final grade, 10 articles remained to be included in the review and one article was also selected from the bibliography of the included study. Thus, a total of 11 studies were included in this systematic review (Fig 1). Eligibility assessment was performed independently in an unblinded standardized manner and disagreements were resolved by discussion.

## Study quality assessment

We assessed the risk of bias, certainty and quality of the included studies with a modified version of the Newcastle–Ottawa Scale [20]. The scale assesses three key points of a given study: participant selection; comparability of the groups; and outcome (s) assessment. We assigned stars for each point of the scale to categorize the studies into good, fair and poor quality based on the NOS [20]. A good quality study was scored 3 or 4 stars in participant selection, 1 or 2 stars in comparability of groups, and 2 or 3 stars in outcome (s) assessment. A fair quality study was scored 2 stars in participant selection, 1 or 2 stars in comparability of the groups, and 2 or 3 stars in outcome (s) assessment. A poor-quality study was scored 0 or 1 star in participant selection, 0 stars in comparability of the groups and 0 or 1 star in outcome (s) assessment (S2 File). In the case of disparity between the two authors (AM and HH) during the study selection process, the disparities were resolved by the decision of the second author (KE).

## Data extraction

We developed a data extraction format based on the Cochrane Consumers and Communication Review Group's data extraction template [21]. The template was pilot-tested on 3 randomly-selected included studies, and refined accordingly. One review author (AM) extracted the data from included studies and the second author (HH) checked the extracted data. Disagreements were resolved by discussion between the two authors and a third author (KE). We extracted information on characteristics of participants such as age range, mean of age, sex ratio and county of origin of refugees and migrant populations. Study characteristics such as first author and year of publication; study location; study population; design; sample size; year (s) data were collected; TB diagnosis method; reported burden of TB (incidence, and prevalence per 100 000 population).

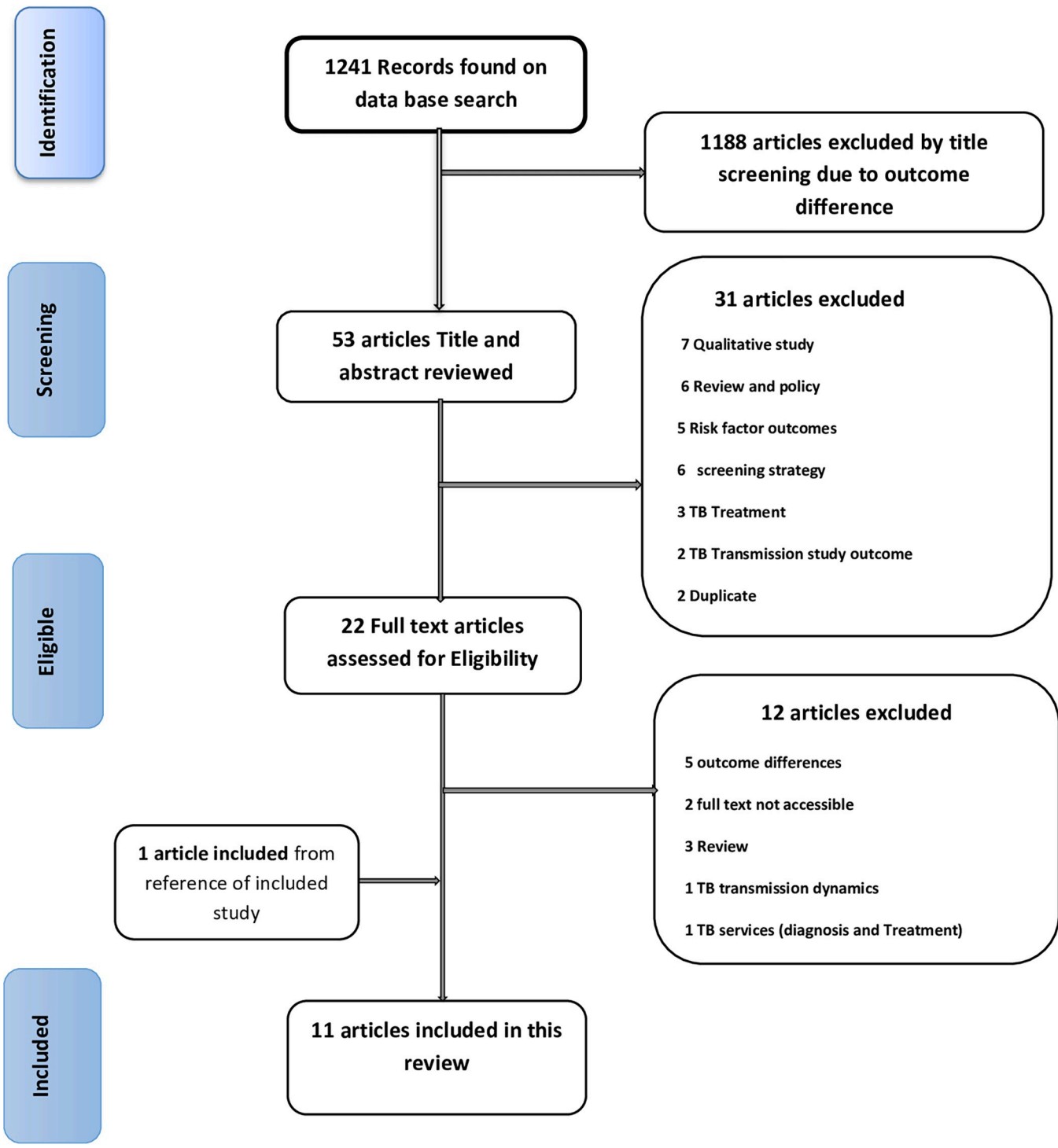

**Fig 1. Selection process flow diagram.**

## Ethics consideration

Ethical approval was not required, as this review was based on previously published articles. However, the protocol of this review was registered on PROSPERO, University of York,

Centre for Reviews and Dissemination and can be accessed with ID number
CRD42020157619 (S3 File).

## Statistical analysis

Extracted data were summarized using Microsoft excel version 16 and exported to STATA version 14 for statistical analysis. Age range, mean age, sex ratio and the most reported countries of origin in TB cases among refugees and migrants were summarized to describe study characteristics. The reported burden of TB (incidence and prevalence) was summarized in a table. TB incidence and/or prevalence were calculated if the number of TB cases and population estimates were provided. Included studies which had reported the above data were eligible for data synthesis. Risk of bias due to missing results were minimized by inclusion of results from sources.

## Results

Elven articles published from studies conducted in 10 countries and addressed TB in migrants, refugees, and asylum seekers. Nine articles were published on refugees and migrants in Europe, three studies in Asia, one study in the United States of America and one in Canada. Half of the included articles (6/11) were published in 2016. The country of origin of refugees and migrants screened for TB varied widely across regions and countries (Fig 2).

Two studies confirmed TB with chest X-Ray (CXR), and three studies confirmed TB diagnosis with bacteriological examination. Both CXR and bacteriological examination used by three studies; and clinical, CXR & bacteriological examination used by two studies as the method of TB diagnosis (Tables 1 and 2). The design and scope of the studies varied across the included studies. Six cross sectional studies, three retrospective cohort studies, one cohort study and one study with a scientific report were included in the present review (Tables 1 and 2). We observed heterogeneity among results of the included studies by grouping the study results in to incidence and prevalence.

More than 3 million refugees and migrants were screened for TB using clinical history of TB, CXR, and bacteriological examination mainly in hospital and reception settings. Ten studies collected data between 2002 and 2017, while one study collected data between 1991 and 2013 (Tables 1 and 2).

Eleven studies have reported the incidence or prevalence of TB in refugees and migrant populations. The incidence of TB ranged from 19 to 754 cases per 100,000 population among the seven included studies (Table 1) and the prevalence of TB ranged from 18.7 to 535 cases per 100,000 population among the four included studies (Table 2).

More than half of the studies had reported TB cases whose age ranges were between 13 to 54 years and males were higher in reported TB cases as compared to females. The most reported countries of origin in TB cases among refugees and migrants were from Asia and Africa. India, Philippines, Pakistan, Syria, Thailand and Vietnam were the most reported countries of origin in TB cases from Asia, whereas Somalia and Eretria were the most reported countries in TB cases from Africa (Table 3).

## Discussion

The current systematic review summarized the magnitude of TB among refugees and migrants. The incidence of TB among refugees and migrants were ranged from 19 to 754 cases per 100,000 population among the seven included studies whereas the prevalence of TB among refugees and migrants were ranged from 18.7 to 535 cases per 100,000 population among the four included studies.

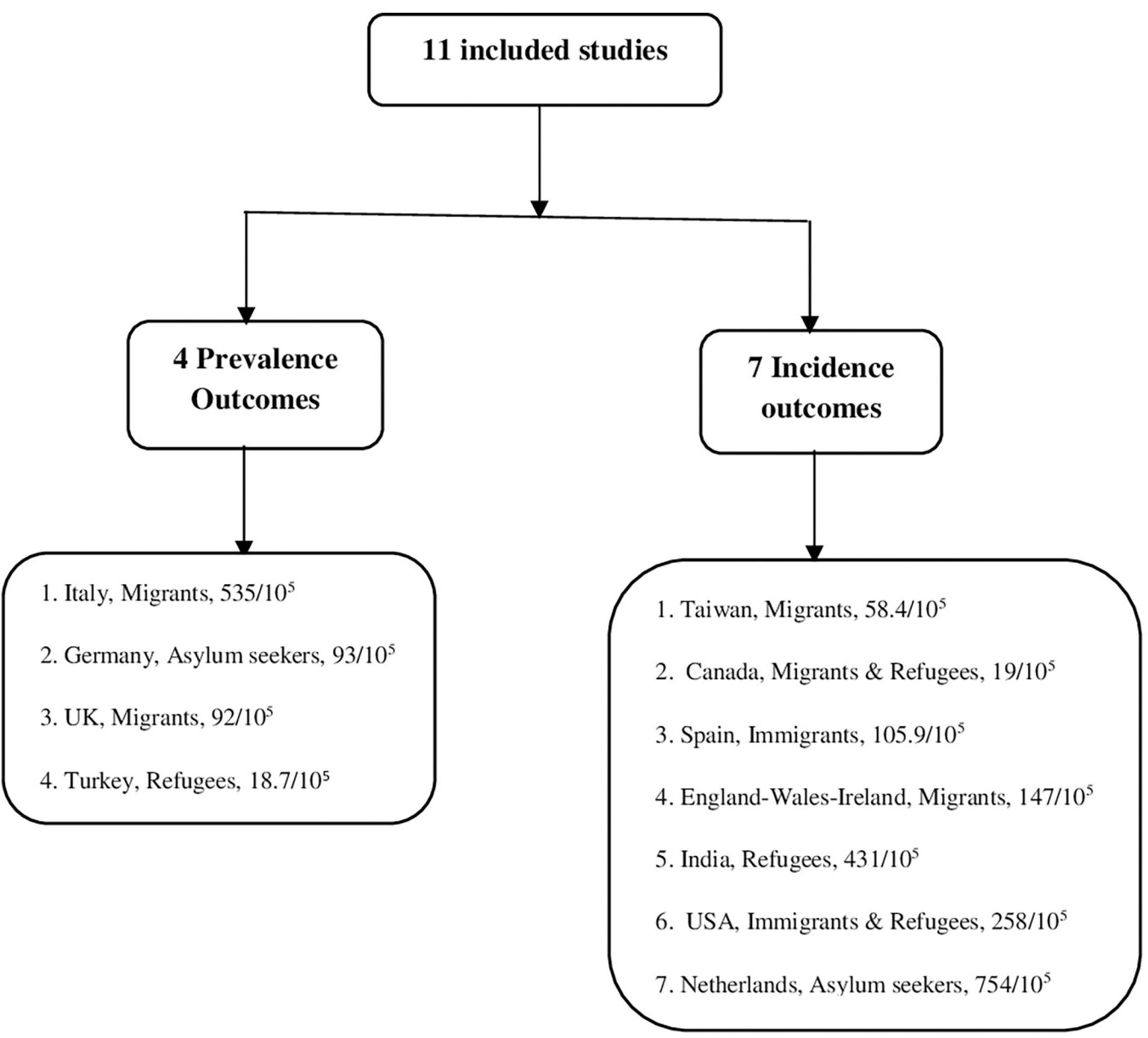

**Fig 2. Results of 11 included studies.**

We found that more articles included in this review were from European countries in the year of publication 2016. This might be due to a large influx of refugees and migrants, mostly from high TB burden countries (HTBCs) of Africa and Asia, who moved mostly to Europe during this period [10, 31]. After the Syrian crisis in 2011, a remarkable increase in TB cases was reported in countries bordering Syria and also affected Europe [30].

The incidence and prevalence of TB among refugees and migrant populations varied among regions and across countries. The lowest incidence (19 cases per 100,000 population) was reported from the study done by Asadi et al., 2017 [22] in Alberta, Canada. This could be due to the study populations' selection, as only 2%(5500) had previous history of TB and it is the fact that previous history of TB is the major determinant in the epidemiology of TB/

**Table 1. Study characteristics and TB Incidence results among the included studies.**

| First author, year of publication | Study location | Design | Study population | Sample size | Year(s) data collected | Type of TB diagnosis | Incidence |
|---|---|---|---|---|---|---|---|
| Lu et al., 2019 [15] | Taiwan | Retrospective Cohort | Migrants | 379422 | 2004 to 2013 | CXR | 58.4 per $10^5$ <br> *Taiwan: 45.5–76.8 per $10^5$* |
| Asadi et al., 2017 [22] | Alberta, Canada | Retrospective cohort | Migrants and refugees | 223225 | 2002 to 2013 | Culture | 19/$10^5$ <br> Alberta: 4.7/$10^5$ |
| Ospinia et al., 2016 [23] | Spain | Cross sectional | Immigrants | 3284 | 1991 to 2013 | CXR and Bacteriologic | 105.9/$10^5$ <br> Highest Pakistan, India, Bangladesh (675/$10^5$), followed by Africa (329/$10^5$) |
| Aldridge et al., 2016 [16] | UK | Cohort | Migrants | 519955 | 2006 to 2012 | CXR and Bacteriologic | 147/$10^5$ |
| Dierberg et al., 2016 [24] | India | Cross sectional | Refugees | 27714 | September 2011 to March 2013 | CXR, Bacteriologic and Xpert MTB/RIF | 431 cases/$10^5$ <br> *India: 181 cases/$10^5$* |
| Liu et al., 2016 [25] | USA | Cross sectional | Immigrants and refugees | 1561460 | 2007 to 2012 | Culture based examination | 258 cases per $10^5$ |
| Boogaard et al, 2020 [26] | Netherlands | retrospective cohort | Asylum seekers | 26,057 | January, 2013, to December 2017 | Clinical, CXR, Bacteriologic | 754 cases per $10^5$ <br> *Netherlands: 4.6/$10^5$* |

MDR-TB [32–34]. However, the reported incidence of migrants was four times higher than the incidence (4.7/100,000 population) overall in the Alberta district in 2014 [22].

The highest incidence cases (754 cases per 100,000 population) were reported from the study done by Boogaard et al., 2020 from Eritrean and Somali refugees in Netherlands [26]. The incidence rate was much higher than the other included studies. A probable explanation for this finding is the additional risk for infection while traveling to Europe, where overcrowding and unsanitary conditions are common along travel routes. The other explanation could be case definition for incident TB case in which both pulmonary and extrapulmonary TB cases were included in the analysis that contribute to the high TB incidence rate in the asylum seekers.

The high incidence cases (431 cases per 100,000 population) were also reported from the study done by Dierberg et al., 2016 from Tibetan refugees in India [24]. This might be due to overcrowded living conditions of Tibetan refugees of whom more than half of TB cases in these refugees occurred in congregated settings. Moreover, use of the rapid diagnostic test and active case finding in the study might increase the detection of undiagnosed TB cases in

**Table 2. Study characteristics and TB prevalence results among the included studies.**

| First author, year of publication | Study location | Design | Study population | Sample size | Year(s) data collected | Type of TB diagnosis | Prevalence |
|---|---|---|---|---|---|---|---|
| Vanino et al., 2017 [27] | Bologna, northern Italy | Cross sectional | Migrants | 3366 | 2014 to 2015 | CXR | 535/$10^5$ <br> *Italy: 6.7/$10^5$* |
| Meir et al., 2016 [28] | Friedland, Germany | Cross sectional | Asylum seekers | 11773 | b/n 2014 and 2015 | Clinical, CXR and Bacteriologic | 93/$10^5$ <br> *Germany: 5.3/$10^5$* |
| Aldridge et al., 2016 [29] | UK | Cross sectional | Migrants | 476455 | between 2005 and 2013 | Bacteriologic confirmation | 92 per $10^5$ |
| Ismail et al, 2018 [30] | Turkey | Scientific report | Refugees | 10689 | ND | ND | 18.7/$10^5$ |

ND-Not Described.

**Table 3. Age range, gender and country of origin of notified TB cases across the included studies.**

| First author, year of publication | Age range of TB cases | Male: Female ratio of TB cases | Country of origin |
|---|---|---|---|
| Lu et al., 2019 [15] | 45–54 | Higher in males (3:1) | Indonesia, Philippines, Thailand, and Vietnam |
| Asadi et al., 2017 [22] | 13–37 | Higher in males | Philippines, India, or China |
| Vanino et al., 2017 [27] | 18–41 | All are males (18/18 males) | West Africa and India |
| Meir et al., 2016 [28] | Mean age 31.2Yrs | ND | Eritrea, Russia, Pakistan and Syria |
| Ospinia et al., 2016 [23] | 25–44 years | Highest in Males (66.7%) | Latin America, Pakistan, India and Bangladesh and Others (Africa and eastern Europe) |
| Aldridge et al., 2016 [16] | 16-44(94.4%) | Higher in Males (67.7%) | 15 HTBCs (Bangladesh, Burkina Faso, Cambodia, Côte d'Ivoire, Eritrea, Ghana, Kenya, Laos, Niger, Pakistan, Somalia, Sudan, Tanzania, Thailand, and Togo) |
| Aldridge et al., 2016 [29] | ≥ 65 | Higher in females 116 (101–134) per $10^5$; Male: 79 (70–90) per $10^5$ | 15 HTBCs (Bangladesh, Burkina Faso, Cambodia, Côte d'Ivoire, Eritrea, Ghana, Kenya, Laos, Niger, Pakistan, Somalia, Sudan, Tanzania, Thailand, and Togo) |
| Dierberg et al., 2016 [24] | ND | Higher in males 77/92 (80.0%) | Tibet |
| Liu et al., 2016 [25] | ND | ND | Vietnam and Philippines |
| Ismail et al, 2018 [30] | ND | ND | Syria |
| Boogaard et al, 2020 [26] | ≥18 | Higher in males (58.7%) | Somalia and Eritrea |

ND-Not Described.

congregate living settings. The reported incidence was more than two times higher compared to the overall incidence in India (181 cases per 100,000 population) in 2010 [24].

The lowest prevalence was reported from Syrian refugees (18.7 cases per 100,000 population) in Turkey [30]. This could be due to the lower TB prevalence reported by both Syria (23/100,000) and Turkey (22/100,000) before the onset of the Syrian crisis, which might result in low TB prevalence of the Syrian refugees in hosting country. After the beginning of the Syrian crisis in 2011, a remarkable increase in TB cases was reported in countries bordering Syria. Turkey is bordered by Syria and reported a noticeable increase in the proportion of imported TB cases from 1.3% in 2011 to 6.8% in 2015 [30].

The highest prevalence of 535 per 100 000 population was reported from asylum seekers in Bologna, northern Italy [27]. The prevalence found in migrants is 80 times greater compared to the estimated prevalence rate in Italy (6.7 per 100 000 population). Europe is the lowest TB burden region and the majority of reports indicated < 20/100,000 population [35]. The highest prevalence in this study compared to other findings in Europe may be explained by the different nature of migration flows and extended staying in detention and overcrowding for several months in Libya before reaching Italian coasts, which might increase the risk of TB transmission and the risk of progression from infection to disease. In contrast, the screening of migrants in a reception setting which underwent by adopting a chest radiograph with a specific TB questionnaire might overestimate the TB screening result, unlike the majority of European countries performing a chest radiograph based on active TB screening [27].

Article search was limited to only three electronic data bases: Google scholar, Medline/PubMed and Web of science. EMBASE and Scopus databases were not accessed due to free access restriction in the institutes where this review conducted.

## Conclusion

The current systematic review showed that the magnitude of TB among refugees and migrant populations was higher than in the host countries in the included studies. The findings revealed the need to implement and improve TB surveillance, promoting TB screening, TB prevention and control in refugees and migrant populations globally.

## Supporting information

**S1 File. PRISMA checklist.**
(PDF)

**S2 File. Study quality assessment.**
(PDF)

**S3 File. PROSPERO registration.**
(PDF)

## Acknowledgments

We thank the Ethiopian Public Health Institute and Aklilu Lemma Institute of Pathobiology, Addis Ababa University.

## Author Contributions

**Conceptualization:** Abyot Meaza, Tedla Mindaye.

**Data curation:** Abyot Meaza, Tedla Mindaye.

**Formal analysis:** Abyot Meaza, Habteyes Hailu Tola, Girmay Medhin.

**Investigation:** Abyot Meaza, Tedla Mindaye.

**Methodology:** Abyot Meaza, Habteyes Hailu Tola, Girmay Medhin.

**Resources:** Tedla Mindaye.

**Supervision:** Balako Gumi.

**Validation:** Abyot Meaza, Kirubel Eshetu, Tedla Mindaye, Balako Gumi.

**Visualization:** Balako Gumi.

**Writing – original draft:** Abyot Meaza.

**Writing – review & editing:** Kirubel Eshetu, Tedla Mindaye, Girmay Medhin, Balako Gumi.

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
