## [Decision Letter · Decision Letter 0]

8 Mar 2022

PONE-D-21-33712

Tuberculosis among refugees and migrant populations: systematic review.

PLOS ONE

Dear Dr. Meaza,

Thank you for submitting your manuscript to PLOS ONE. After careful consideration, we feel that it has merit but does not fully meet PLOS ONE’s publication criteria as it currently stands. Therefore, we invite you to submit a revised version of the manuscript that addresses the points raised during the review process.

We look forward to receiving your revised manuscript.

Kind regards,

Felix Bongomin, MB ChB, MSc, FECMM

Academic Editor

PLOS ONE

“The author(s) declare that there is no any conflict of interest that might influence this work”

5. We noticed you have some minor occurrence of overlapping text with the following previous publication(s), which needs to be addressed:

- https://www.iosrjournals.org/iosr-jdms/papers/Vol14-issue5/Version-3/M014535862.pdf

- https://www.dovepress.com/identification-of-mycolic-acid-forms-using-surface-enhanced-raman-scat-peer-reviewed-fulltext-article-IJN

- https://link.springer.com/chapter/10.1007/978-1-4419-0748-6_24

- http://apps.who.int/iris/bitstream/handle/10665/208780/RS_2013_GE_13_PHL_eng.pdf;jsessionid=7B1E642A119B2E221EED737065121CAC?sequence=1

The text that needs to be addressed involves the Introduction. 

In your revision ensure you cite all your sources (including your own works), and quote or rephrase any duplicated text outside the methods section. Further consideration is dependent on these concerns being addressed.

Additional Editor Comments:

The reviewer have raised important comments that needs to be addressed prior to editorial decision on your manuscript.

Reviewers' comments:

Reviewer's Responses to Questions

**Comments to the Author**

1. Is the manuscript technically sound, and do the data support the conclusions?

Reviewer #1: Partly

2. Has the statistical analysis been performed appropriately and rigorously? 

Reviewer #1: No

3. Have the authors made all data underlying the findings in their manuscript fully available?

Reviewer #1: Yes

4. Is the manuscript presented in an intelligible fashion and written in standard English?

Reviewer #1: No

5. Review Comments to the Author

Reviewer #1: Dear author

The authors presented systematic review aimed to summarize the evidence on the magnitude of TB among refugees and migrant populations. My comments were supplied in the PDF file of the manuscript. My decision is major revisions.

6. PLOS authors have the option to publish the peer review history of their article (what does this mean?). If published, this will include your full peer review and any attached files.

Reviewer #1: **Yes: **Marwa I Abd El-Hamid

---

## [Author Response · Author response to Decision Letter 0]

3 Apr 2022

Dear PLOS ONE Editorial Office,

I would like to thank the editorial office for the suggestions for correction and reviewer for comments for revision. This is to request your esteemed office for review and publication of my revised manuscript entitled “Tuberculosis among refugees and migrant populations: systematic review.”

Points of revision and responses for reviewer’s comments mentioned below: 

The manuscript revised based on PLOS ONE's style requirements and templates (see the revised manuscript, and the marked manuscript. Files saves as Fig 1 and 2. tiff.

2. As per your suggestion, I would like to remind the editorial office my amendment statement to competing interest in order to change the online submission form on my behalf.

Competing interest: “The author(s) declare that there is no any conflict of interest that might influence this work. This does not alter our adherence to PLOS ONE policies on sharing data and materials”. 

3. As per your recommendation, I would like to remind the editorial office my amendment statement to data availability statement in order to change the online submission form on my behalf.

Data Availability statement: “All data and materials that support the final results are presented in the manuscript and its supporting file”. 

4. Please include captions for your Supporting Information files at the end of your manuscript, and update any in-text citations to match accordingly.

Captions inserted for supporting information (PRISMA checklist).

5. We noticed you have some minor occurrence of overlapping text with the following previous publication(s), which needs to be addressed:

The text that needs to be addressed involves the Introduction. In your revision ensure you cite all your sources (including your own works), and quote or rephrase any duplicated text outside the methods section. Further consideration is dependent on these concerns being addressed.

The manuscript ensured from duplicate texts. These has been shown in track changes in the marked manuscript. 

6 While revising your submission, please upload your figure files to the Preflight Analysis and Conversion Engine (PACE) digital diagnostic tool.

Figure files (Fig. 1 and 2) uploaded to the PACE tool and the figures meets the PACE tool format.

Response for Reviewer’s comment

The authors presented systematic review aimed to summarize the evidence on the magnitude of TB among refugees and migrant populations. My comments were supplied in the PDF file of the manuscript. My decision is major revisions.

We believe the comments help the quality of the manuscript. Therefore, all comments are accepted and submitted in revised versions (see Manuscript and Manuscript marked). Moreover, our response for the comments listed below in table.

Section Comments Response

Abstract Please rewrite the aim in brief Accepted and revised as indicated on page 2, in line 26-30

Abstract Why did the authors only talked the incidence in the abstract We presented incidence and prevalence as key findings and now we added additional finding as indicated on page 2 in line 38-39 and 41-44.

Key words Add more immigrants, asylum-seekers, and refugee-camps added as additional key words on page 3, line 51-52.

Introduction Add references to this sentence Four additional references (reference nr 13-16) added to support the stated statement as indicated on page 5, line 80. 

Methods, search strategy Transfer to authors contribution Removed from page 6, line 105-106 and transferred to page 15, line 255-256.

Methods, Study selection process Transfer to authors contribution Removed from page 6, line 119-120 and transferred to page 15, line 256-257.

Statistical Analysis Please mention the program used Thank you for the comment and mentioned on page 8, 148-149.

 Results Please provide the results with figures illustrating with results Analysis Accepted and provided as figure 2.

Results Please refer also to diagnosis and control of TB As per the included studies, the data supports to summarize the method of TB diagnosis and this has been summarized as indicated on page 9, 164-167 in addition to, in Table 1 and 2.

Results Tables not table Thank you for the comment and corrected on page 9, line 169 and page 11, line 178.

Discussion Explain The reason for the discussion point explained and supporting facts added on page 13, line 205-207 and reverence nr 33-35 cited

Conclusion Write a conclusion for the study The conclusion made as the end paragraph of discussion and now transferred as separate conclusion as indicated on page 15, line 246-51

References Add more recent references Accepted and 7 recent references added (reference cited from 13-16, and 33-35) as indicated on page 18 and page 20-21.

---

## [Editor Report · Decision Letter 1]

6 May 2022

Tuberculosis among refugees and migrant populations: systematic review.

PONE-D-21-33712R1

Dear Dr. Meaza,

We’re pleased to inform you that your manuscript has been judged scientifically suitable for publication and will be formally accepted for publication once it meets all outstanding technical requirements.

Kind regards,

Felix Bongomin, MB ChB, MSc, MMed, FECMM

Academic Editor

PLOS ONE
---

## [Editor Report · Acceptance letter]

26 May 2022

PONE-D-21-33712R1 

Tuberculosis among refugees and migrant populations: systematic review. 

Dear Dr. Meaza:

I'm pleased to inform you that your manuscript has been deemed suitable for publication in PLOS ONE. Congratulations! Your manuscript is now with our production department. 

Kind regards, 

on behalf of

Dr. Felix Bongomin 

Academic Editor

PLOS ONE